# Investigation of the Relationship between Right Coronary Artery–Aorta Angle and Coronary Artery Disease and Associated Risk Factors

**DOI:** 10.3390/jcm12031051

**Published:** 2023-01-29

**Authors:** Jade Geerlings-Batt, Ashu Gupta, Zhonghua Sun

**Affiliations:** 1Discipline of Medical Radiation Science, Curtin Medical School, Curtin University, Perth, WA 6845, Australia; 2Medical Imaging Department, Fiona Stanley Hospital, Perth, WA 6150, Australia; 3Curtin Medical School, Faculty of Health Sciences, Curtin University, Perth, WA 6845, Australia

**Keywords:** coronary computed tomography angiography, coronary artery disease, disease risk factors, right coronary artery, right coronary artery-aorta junction, atherosclerosis

## Abstract

At the level of the left coronary artery tree, there is evidence showing an association between bifurcation angle and coronary artery disease (CAD), and this motivated us to explore similar associations at the level of the right coronary artery (RCA). The purpose of this study was to determine whether there is a relationship between RCA–aorta angle and CAD and age, sex, body mass index, smoking status, hypertension, and high blood cholesterol. The coronary computed tomography angiography datasets and CAD risk factor checklists of 250 patients were retrospectively reviewed, with RCA–aorta angles measured via multiplanar reformation images. Independent *t*-tests were used to compare mean RCA–aorta angle measurements between groups, correlations between continuous variables were assessed using Pearson and Spearman correlations, and a general linear model was used to adjust for potentially confounding variables. Coronary angle measurements were conducted by two independent assessors with very strong intraclass correlation (r=0.999, p<0.001). A significantly smaller mean RCA–aorta angle was observed in the CAD group (79.07 ± 24.88°) compared to the normal group (92.08 ± 19.51°, p=0.001), in smokers (76.63  ± 22.94°) compared to non-smokers (85.25  ± 23.84°, p=0.016), and a narrow RCA–aorta angle was negatively correlated with BMI (r=−0.174, p=0.010). This study suggests a relationship between narrow RCA–aorta angles and CAD, smoking, and increasing BMI.

## 1. Introduction

Coronary artery disease (CAD) is characterised by atherosclerotic plaque formation within the coronary arteries and is the most common type of cardiovascular disease, affecting an estimated 5.7% of the Australian adult population in 2018 [1]. CAD typically involves progressive luminal narrowing and the development of coronary stenosis, which can inhibit adequate coronary circulation and prompt the onset of associated conditions, such as those comprising acute coronary syndrome [2,3]. CAD, left undiagnosed and untreated, has the potential to significantly reduce a person’s life expectancy, as well as their wellbeing and quality of life [3]. Hence, properly understanding CAD pathogenesis is essential in facilitating the identification of at-risk individuals, providing earlier, more effective treatment, and ultimately improving patient outcomes.

The left coronary artery (LCA) arises from the aorta and bifurcates into the left anterior descending (LAD) and left circumflex (LCx) arteries, and several recent studies have identified an association between wide LCA bifurcation (LAD–LCx) angles and CAD (Figure 1) [4,5,6,7,8]. Haemodynamics have since been analysed using computational fluid dynamics (CFD), which demonstrated significantly reduced wall shear stress (WSS), as well as larger areas of reduced WSS in wide (>80°), non-stenosed bifurcations (Figure 2) [9,10,11,12]. This has led investigators to theorise a direction of causality, suggesting WSS may play a protective role against atherosclerosis development, with altered intraluminal forces and reduced WSS predisposing coronary vessels to progressive plaque deposition [9,10,11,12]. Hence, individuals with a wide LAD–LCx angle may be more likely to develop CAD than those with a narrow angle, and a number of studies have confirmed these associations by comparing coronary computed tomography angiography (CCTA) findings with invasive coronary angiography (ICA) [4,13,14,15,16,17].

The LAD–LCx angle has been studied from both causal-comparative and CFD perspectives, with evidence showing the associations between the LAD–LCx angle and CAD [13,14,15,16,17]. This motivated us to explore the relationships between the right coronary artery–aorta (RCA–aorta) angle and CAD. The findings of a pilot study by Geerlings-Batt and Sun indicated a possible association between the RCA–aorta angle and CAD [18]. However, their sample was small, consisting of 30 normal and 30 CAD cases, with the authors suggesting their method be replicated as part of a larger study to validate their preliminary findings. Hence, the purpose of this study is to determine whether there is a relationship between the RCA–aorta angle and CAD, and we hypothesise that CAD is associated with narrow RCA–aorta angles.

Additionally, age, sex, body mass index (BMI), tobacco smoking status, diabetes, hypertension, and high blood cholesterol are well-documented characteristics known to increase a person’s likelihood of developing CAD [1,3]. The interactions between these behavioural and biochemical risk factors and CAD have been widely investigated, and information regarding these characteristics is often routinely collected from patients presenting for CCTA. This study also aims to identify relationships between CAD risk factors, age, sex, BMI, smoking status, diabetes, hypertension, and high blood cholesterol, and RCA–aorta and LAD–LCx angles, to further characterise the role variable coronary anatomy plays in CAD development.

## 2. Materials and Methods

### 2.1. Sampling and Data Collection

The medical records of 342 consecutive patients (patients presenting with angina pectoris, with or without a family history of CAD, and/or those with abnormal echocardiogram or electrocardiogram results) who underwent CCTA for suspected CAD between January–May 2022 at a major public hospital were screened for inclusion and retrospectively reviewed. Only CCTA cases with completed contrast-enhanced coronary angiography and calcium scoring were included in this study. As this study aimed to examine naturally occurring relationships between coronary angles and CAD and associated risk factors, 16 patients with a history of coronary artery bypass grafting and/or stenting were excluded. Two patients with metallic implants (prosthetic valves, stents, pacemakers, etc.) were also excluded due to suboptimal coronary artery visualisation caused by streak artefacts. Additionally, 48 patients were excluded due to illegible, incomplete, or absent CAD risk factor checklists, 7 did not undergo calcium scoring, and 9 patients exhibited at least one anatomical variation preventing coronary angle measurement and were not included in this study. Coronary angles could not be accurately measured for 10 cases due to poor image quality caused by either motion artefacts or photon starvation, and these patients were consequently also excluded. A total of 250 patients met our selection criteria and were included in this study.

Non-anonymised CCTAs, calcium scores and patient risk factor checklists were accessed via the picture archiving and communication system (PACS) on a workstation at the clinical site. CAD risk factors, age, sex, type 2 diabetes (yes or no), current smoker status (yes or no), hypertension (yes or no), and high blood-cholesterol (yes or no), were recorded for each patient, with BMI (kg/m^2^) calculated from their documented height and weight. RCA–aorta and LAD–LCx angle measurements were also acquired at the workstation, and these data were anonymised for analysis off-site. This study was approved by the Curtin University Human Research Ethics Committee and the clinical centre. All procedures performed in this study aligned with the 1964 Declaration of Helsinki and were undertaken in accordance with the ethical standards of the involved institutions. Informed consent was not provided by patients, given this study’s retrospective design.

### 2.2. CCTA Scanning Protocols and Image Reconstruction

CCTA scans were acquired on a 256-slice Siemens SOMATOM Definition Flash CT scanner (SOMATOM Definition Flash, Siemens, Erlangen, Germany). Oral beta blockers were administered for all patients with a heart rate above 65 beats per minute (bpm) to minimise image degradation by motion artefact. Patients with heart rates <65 bpm and >65 bpm were imaged using prospective ECG gating according to the institutional FLASH and adaptive sequential protocols, respectively. All patients were administered 70 mL (not inclusive of test boluses) of a non-ionic intravenous iodinated contrast medium, such as Iohexol, followed by a saline chaser, and CCTAs were acquired using either a test bolus or bolus-tracking technique. Imaging was performed using a tube voltage of ~120 kVp, with tube current selected by the automatic dose modulator for each patient. Axial images were reconstructed with a slice thickness of 0.75 mm at 0.6 mm intervals, and multiplanar reformatted images (MPRs) were created from these axials with variable obliquity, depending on patient anatomy.

### 2.3. Coronary Angle Measurement

Two-dimensional (2D) axial and multiplanar reformation (MPR) images were used to perform LAD–LCx angle measurements, as the accuracy of these methods is comparable to using three-dimensional (3D) views [4]. However, our recent study determined using axials alone to be a reliable, yet highly inaccurate method of measuring the RCA–aorta angle, as the accuracy of this approach is dependent on RCA trajectory [18]. If the RCA follows an immediately inferior path, an axial-acquired RCA–aorta measurement will be larger than the true angle size. Consequently, this method generally results in a consistently overestimated RCA–aorta angle (Figure 3), and it was concluded that this angle may be better assessed using MPRs [18]. Hence, the RCA–aorta angle was measured via MPR images in this study, by manipulating obliqued planes to identify and measure the smallest observable RCA–aorta angle. Each LAD–LCx and RCA–aorta angle was measured three times, with the mean measurement used to limit measurement error. Twenty-five cases were randomly selected for inter-observer variability testing by two independent assessors.

### 2.4. Statistical Analysis

SPSS (version 28, IBM SPSS Statistics, New York, NY, USA) was used for data analysis, with continuous variables expressed as mean ± standard deviation, and categorical variables expressed as frequencies and percentages. All continuous variables were tested for normality, and independent *t*-tests were used to assess for significant mean coronary angle differences between patients grouped according to CAD, sex, smoking status, diabetes, blood pressure, and blood cholesterol. LAD–LCx and RCA–aorta angles were each correlated with age and BMI, as well as with each other using Pearson correlation. The LAD–LCx and RCA–aorta angles from CAD cases were also correlated with calcium score using Spearman correlation, due to its non-normal distribution. A general linear model (GLM) was used to adjust for possible confounding interactions between CAD and several risk factors, age, sex, and BMI, following *t*-test analysis. Inter-observer agreeance was determined via an intraclass correlation coefficient. A *p*-value < 0.05 indicated statistical significance.

## 3. Results

The CCTA cases and medical records from 250 patients (91 females, 159 males, mean age 57 ± 11.12) were included in this study, with patient characteristics summarised in Table 1. Age, BMI, and LAD–LCx and RCA–aorta angles were normally distributed, whereas the calcium score was non-normally distributed. The mean LAD–LCx angle of the sample was 75.07 ± 29.15°, ranging from 17.43° to 158.67°, and the mean RCA–aorta angle was 99.94  ± 27.67°, ranging from 10.83° to 129.2°. The RCA–aorta and LAD–LCx angles were measured by two independent assessors via MPR images, and axial and MPR images, respectively, with very strong intraclass correlation (r=0.998−0.999,p<0.001).

As per independent *t*-test analysis (Table 2), a significantly larger mean LAD–LCx angle (p=0.002) and a significantly smaller mean RCA–aorta angle (p<0.002) were observed in the CAD group compared to the normal group (Figure 4). Males had a significantly larger mean LAD–LCx angle (p=0.004) and a significantly smaller mean RCA–aorta angle (p=0.016) compared to females. Patients with diabetes had a significantly larger LAD–LCx angle compared to nondiabetic patients (p=0.037), and patients who smoked had a significantly smaller RCA–aorta angle compared to those who did not smoke (p=0.016). There were no statistically significant associations between the LAD–LCx angle and smoking (p=0.750), hypertension (p=0.949), or high blood-cholesterol (p=0.694), or between the RCA–aorta angle and diabetes (p=0.110), hypertension (0.246), or high blood cholesterol (0.218).

There was a significant weak positive correlation between the LAD–LCx angle and BMI (r=0.138, p=0.030) and a significant weak negative correlation between the RCA–aorta angle and BMI (r=−0.229, p<0.001). However, no significant relationships were identified between the LAD–LCx and RCA–aorta angles and age (p=0.873 and p=0.771, respectively) or between the LAD–LCx and RCA–aorta angles and calcium score (p=0.925 and p=0.852, respectively) (Table 3). The GLM (Table 4) revealed significant associations between wide LAD–LCx angles and CAD (p=0.033) and males (p=0.024), as well as between narrow RCA–aorta angles and CAD (p=0.001) and BMI (r=−0.174, p=0.010). However, the GLM did not yield statistically significant results for relationships between the LAD–LCx angle and age (p=0.874) or BMI (p=0.101), or between the RCA–aorta angle and age (p=0.553) or sex (p=0.140). The GLMs also included CAD * sex, which returned no significant associations with the LAD–LCx (p=0.771) or RCA–aorta (p=0.321) angles.

## 4. Discussion

### 4.1. Relationship between RCA–Aorta Angle and CAD

The results of this study suggest a relationship between narrow RCA–aorta angles and CAD (Figure 5), supporting the preliminary findings of our recent study [18]. There is currently very little literature available discussing possible relationships between the RCA–aorta angle and CAD. However, one 1984 post-mortem study was conducted with the purpose of determining whether the coronary take-off angle and the presence of ostial valve-like ridges were related to sudden death [19]. This study involved the dissection and analysis of 41 cadaveric hearts, with the findings suggesting an association between narrow coronary take-off (RCA–aorta and LCA-aorta) angles and sudden death, even in the absence of atherosclerosis [19]. The authors postulated that aortic root dilation (ARD) may play a role in compressing the coronary arteries and impairing blood flow, consequently increasing the risk of an acute cardiac event [19]. This impaired blood flow could also affect CAD development. Whereas the definition of ‘sudden death’ is relatively non-specific, these findings were subsequently supported [20]. The coronary take-off angle has since been investigated in conjunction with take-off height, only providing evidence to suggest narrow angles may impair blood flow within high take-off coronary arteries, specifically [20]. The effect of narrow RCA–aorta angles on CAD development is yet to be examined in isolation. If both the RCA–aorta angle and height are related to CAD development, future analyses should account for possible interactions between these variables, and CFD should be conducted to identify resultant changes in intraluminal forces. Providing the role WSS plays in preventing CAD development is universal, atherosclerotic plaque formation in patients with narrow RCA–aorta angles could be attributed to reduced WSS [9,10,11,12]. An identical relationship may exist contralaterally, and these studies could be replicated to determine how LCA–aorta characteristics may affect CAD development.

Additional studies may also correlate the RCA–aorta angle with the degree of coronary stenosis, plaque location, and plaque type (calcified, non-calcified or mixed) to gain further insight into the relationship between the RCA–aorta angle and CAD. However, studies investigating possible correlations between the RCA–aorta angle and the degree of coronary stenosis would require the utilisation of ICA, as blooming artefacts caused by partial volume averaging tend to worsen the appearance of significant stenosis on CCTA [21,22,23]. Alternatively, the degree of coronary stenosis expressed as the extent of occlusion does not always perfectly correspond to physiological deterioration, and future studies could instead use CT-derived fractional flow reserve (FFRCT) to better understand how the RCA–aorta angle may be related to ischaemic severity [24,25].

### 4.2. RCA–Aorta Angle and CAD Risk Factors

There is currently no evidence available explicitly describing relationships between CAD risk factors and the RCA–aorta angle. However, if ARD is involved in RCA compression, possible relationships between the RCA–aorta angle and CAD risk factors can be theorised. To an extent, the aortic root dilates naturally with age as elastic and collagen fibres degenerate [26]. However, progressive, accelerated ARD is typically associated with smoking [26,27], hypertension, and various inflammatory diseases [26]. This may explain the apparent relationship between narrow RCA–aorta angles and smoking but does not account for the lack of a relationship between narrow RCA–aorta angles and hypertension in our study. Additionally, diabetes plays a poorly understood protective role against ARD and aortic aneurysm, and Miyama et al. concluded that hyperglycaemia may cause reduced macrophage infiltration and matrix metalloproteinase-9 (MMP9) levels, both of which are involved in promoting aneurysm and possibly ARD development [28]. This aligns with our results, which do not suggest an association between diabetes and narrow RCA–aorta angles.

Although sex was related to the RCA–aorta angle according to independent *t*-test analysis, this relationship was not significant using the GLM. This suggests either that our initial results may have been influenced by confounding variables, or statistical power is limited, and the sample size is too small for a GLM to return statistically significant results. The results of this study also indicated a significant weak negative correlation between BMI and the RCA–aorta angle, suggesting increased BMI may be related to narrow RCA–aorta angles. However, there is little evidence available to explain this. Increased relative pericardial fat volume [29] and left [30] and right ventricular hypertrophy [31] are both associated with high BMI and could theoretically affect coronary artery trajectory. However, further research is required to determine whether and how various anatomical changes associated with increased BMI may alter the RCA–aorta angle. 

### 4.3. Relationship between LAD–LCx Angle and CAD

This study’s CAD group exhibited a significantly larger mean LAD–LCx angle compared to the normal group, providing further evidence to support the relationship between wide LCA bifurcation angles and CAD development. This relationship has been identified by several other studies employing similar methods [16]. Additionally, multiple independent CFD analyses have demonstrated a correlation between the LAD–LCx angle and variable WSS [9,10,12]. This allowed investigators to infer a probable direction of causality, with altered haemodynamics observed in non-stenosed, wide LCA bifurcations suggesting a predisposition to developing atherosclerotic plaques. Our investigation of the LAD–LCx angle only analysed its relationship with the general presence of CAD; hence, future studies should aim to correlate both coronary angles with the degree of coronary stenosis, plaque location, and plaque type. Although some of the literature has discussed possible relationships relating to these variables, the strength of current evidence is limited by the low number of studies, contradicting results, and the use of potentially invalid methods [14]. Consequently, whether a definite relationship exists between the LAD–LCx angle and the degree of coronary stenosis and the plaque type and location remains indeterminable. A recent systematic review of 13 studies highlighted the reduced validity of these studies reporting the relationship between the LAD–LCx angle and CAD, and further exploration of the correlation between the LAD–LCx angle and patient outcomes is needed to reinforce the current evidence [14].

### 4.4. LAD–LCx Angle and CAD Risk Factors

According to the results of this study, the LAD–LCx angle may be related to sex and diabetes, with males and individuals with diabetes exhibiting a greater mean LAD–LCx angle than females and non-diabetics, respectively. Additionally, the LAD–LCx angle was not related to age, smoking status, hypertension, or high blood cholesterol. Of the few studies analysing relationships between the LAD–LCx angle and CAD risk factors, Temov et al. also found the LAD–LCx angle to be associated with sex, calculating males to be 2.07 times more likely to have an LCA bifurcation angle >80° [15]. This has been attributed to generally larger body habitus resulting in expanded coronary angles amongst males [15]. Relative pericardial fat volume tends to be greater in males than females [32] and is known to increase with BMI [29]. Hence, larger volumes of pericardial fat may also affect the LCA bifurcation angle; however, this requires further investigation.

Additionally, Temov et al. also determined that LAD–LCx was associated with BMI and was unrelated to diabetes [15], which may be due to differences in statistical analysis. For example, the LAD–LCx angle was categorical (cases were organised into LAD–LCx < 80° and LAD–LCx > 80°), and a GLM was not utilised by Temov et al. [15]. Hence, either their study was affected by random differences in their sample, by confounding variables, or our study lacks the sample size to achieve statistical significance using a GLM. Cui et al. also categorised LAD–LCx according to a cut-off angle of 78° and found the LCA–bifurcation angle to be unrelated to BMI or diabetes [16], creating further incongruity. The current literature is inconsistent, and there is currently insufficient evidence available to definitively ascertain whether the LAD–LCx angle is related to BMI or diabetes.

### 4.5. Study Limitations and Directions for Future Research

This study did not correlate coronary angles with degrees of stenosis, plaque location, or plaque type, and performing an accurate GLM incorporating more independent variables was impossible with the given sample size. Future research should use ICA- or CCTA-derived data relating to disease severity to ascertain whether this is related to the RCA–aorta or LAD–LCx angle. Subsequent studies with larger samples are required to perform a more accurate GLM and properly adjust for confounders. Factors potentially causing more acute RCA–aorta angles, such as ARD, should also be investigated. Additional research is required to determine if ARD, the RCA–aorta angle, and RCA–aorta height might interact in predisposing individuals to CAD development. Future causal-comparative and CFD studies should analyse angles in conjunction with height to determine how these variables are correlated with CAD and changing intraluminal forces, respectively. CFD analysis is undoubtedly the next step in understanding plaque progression in narrow RCA–aorta bifurcations. Additionally, our method of measuring the RCA–aorta angle may prove time-consuming for clinicians, potentially limiting workflow efficiency. This could make our method unappealing for routine implementation in its current state. Future studies should explore more time-effective ways of accurately measuring RCA–aorta angles, such as via automated tools and software.

Given the nature of CAD as our primary independent variable, as well as the use of ionising radiation in performing CCTA, it was impossible to randomise our sampling. Bias was consequently inherent and unavoidable, due to the employment of purposive sampling and the exclusion of incomplete data. As risk factor data were reported by patients via their pre-CCTA risk factor checklists, this also creates the potential for self-reporting bias. Hence, this may have reduced the extent to which our sample reflected the larger population. Future studies should aim to negate these biases, such as by including incomplete data and utilising more comprehensive data analyses to overcome associated challenges.

## 5. Conclusions

Our findings suggest a relationship between narrow RCA–aorta angles and CAD, sex, and support the previously documented relationships between the LAD–LCx angle and CAD and sex. However, relationships between coronary angles and other CAD risk factors, BMI, smoking status, and diabetes, remain uncertain and require further investigation. Future research should also incorporate larger sample sizes to achieve more accurate GLM-, ICA-, or FFRCT-derived data to assess relationships with CAD severity and CFD to begin understanding the direction of causality between the RCA–aorta angle and CAD. Very little is known regarding factors affecting the RCA–aorta angle itself, and studies investigating possible relationships with ARD and RCA–aorta height are warranted to better understand the cause of suboptimal coronary anatomy and potentially mitigate CAD progression.

## Figures and Tables

**Figure 1 jcm-12-01051-f001:**
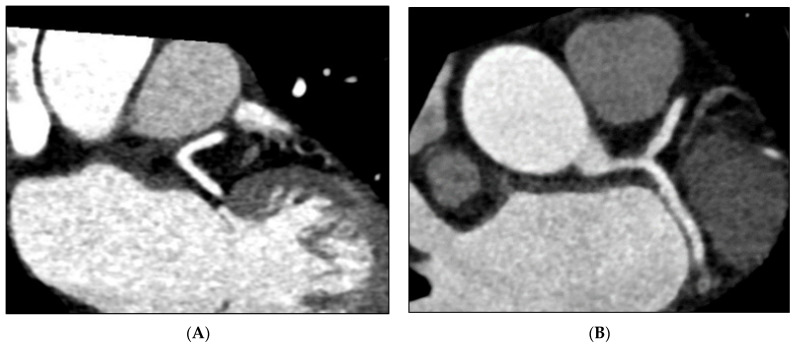
Multiplanar reformation (MPR) images depicting a narrow LAD–LCx angle from an individual without CAD (**A**) and a wide LAD–LCx angle from an individual with CAD (**B**).

**Figure 2 jcm-12-01051-f002:**
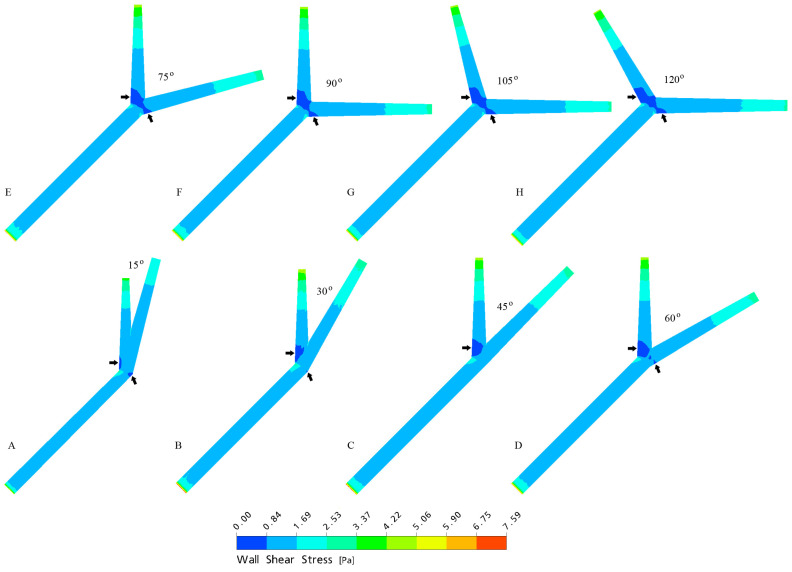
Models derived from CFD analysis simulating LAD–LCx angles during the peak systolic phase (**A**–**H**), demonstrating reduced WSS and an enlarged area of reduced WSS (arrows) in wide bifurcation angles. Reproduced with permission from Chaichana et al. [9].

**Figure 3 jcm-12-01051-f003:**
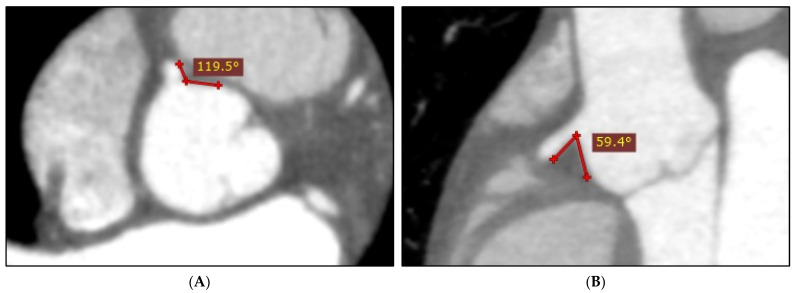
Axially acquired RCA–aorta measurement (**A**) demonstrating overestimation of the true angle size (**B**).

**Figure 4 jcm-12-01051-f004:**
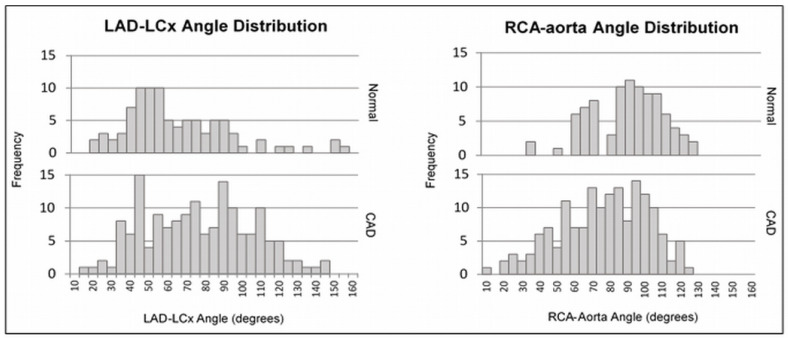
LAD–LCx and RCA–aorta angle distributions for normal and CAD groups.

**Figure 5 jcm-12-01051-f005:**
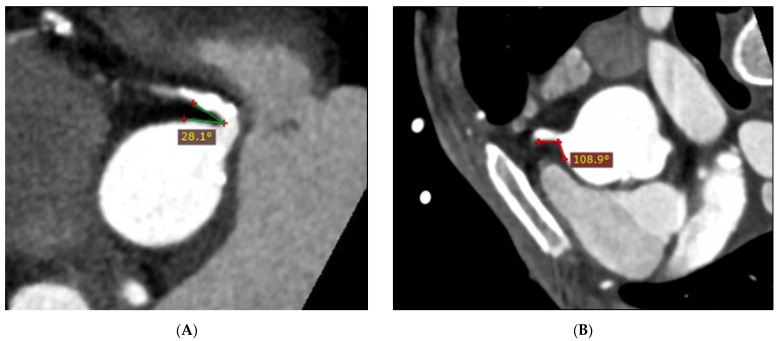
MPR images of a narrow (28.1°) RCA–aorta angle from an individual with CAD (**A**) and calcified plaques, and a wide (108.9°) RCA–aorta angle from a normal case (**B**).

**Table 1 jcm-12-01051-t001:** Demographic characteristics of the study patients.

Variables		Normal (*n* = 91)	CAD (*n* = 159)
Age		52.56 ± 11.27	60.25 ± 10.04
BMI		28.24 ± 6.40	30.83 ± 6.94
LAD–LCx angle		67.48 ± 28.73°	79.39 ± 28.58°
RCA–aorta angle		92.08 ± 19.51°	79.07 ± 24.88°
Sex	Female	54 (59.34%)	66 (41.51%)
	Male	37 (40.66%)	93 (58.49%)
Smoking status	Non-smoker	75 (82.42%)	133 (83.65%)
	Smoker	16 (17.58%)	26 (16.35%)
Diabetes	Non-diabetic	80 (87.91%)	119 (74.84%)
	Diabetic	11 (12.09%)	40 (25.16%)
Blood pressure	Normal	53 (58.24%)	51 (32.08%)
	Hypertensive	38 (41.76%)	108 (67.92%)
Blood cholesterol	Normal	67 (73.63%)	80 (50.31%)
	High blood cholesterol	24 (26.37%)	79 (49.69%)

CAD: coronary artery disease, LAD: left anterior descending, LCx: left circumflex, RCA: right coronary artery, BMI: body mass index.

**Table 2 jcm-12-01051-t002:** Summary of independent *t*-test results between coronary angle measurements and risk factors.

Dependent	Groups	Mean ± Standard Deviation	*p*-Value
LAD–LCx	Normal	67.49 ± 28.73	0.002
	CAD	79.39 ± 28.57	
LAD–LCx	Female	69.54 ± 28.48	0.004
	Male	80.15 ± 58.93	
LAD–LCx	Non-smoker	75.36 ± 28.09	0.750
	Smoker	73.58 ± 34.24	
LAD–LCx	Non-diabetic	73.04 ± 28.65	0.037
	Diabetic	82.92 ± 30.00	
LAD–LCx	Normal	75.20 ± 30.54	0.949
	Hypertensive	74.95 ± 28.22	
LAD–LCx	Normal	74.46 ± 29.95	0.694
	High cholesterol	75.92 ± 28.07	
RCA–aorta	Normal	92.08 ± 19.51	<0.001
	CAD	79.07 ± 24.88	
RCA–aorta	Female	87.15 ± 23.64	0.016
	Male	80.71 ± 23.75	
RCA–aorta	Non-smoker	85.25 ± 23.84	0.016
	Smoker	76.63 ± 22.94	
RCA–aorta	Non-diabetic	84.79 ± 23.50	0.110
	Diabetic	79.96 ± 25.12	
RCA–aorta	Normal	84.99 ±20.97	0.246
	Hypertensive	82.95 ± 25.77	
RCA–aorta	Normal	84.80 ± 23.16	0.218
	High blood-cholesterol	82.37 ± 24.89	

Abbreviations same as in Table 1.

**Table 3 jcm-12-01051-t003:** Summary of Pearson and Spearman correlation results.

Dependent	Independent	Coefficient	*p*-Value
LAD–LCx	Age	0.010	0.873
LAD–LCx	BMI	0.138	0.030
LAD–LCx	Calcium score	0.008	0.925
RCA–aorta	Age	0.018	0.771
RCA–aorta	BMI	−0.229	<0.001
RCA–aorta	Calcium score	0.012	0.852
RCA–aorta	LAD–LCx	−0.053	0.407

Abbreviations same as in Table 1.

**Table 4 jcm-12-01051-t004:** General linear model outputs.

Dependent	Independents	*p*-Value
LAD–LCx	CAD	0.033
	Sex	0.024
	Age	0.874
	BMI	0.101
	CAD * Sex	0.771
RCA–aorta	CAD	0.001
	Sex	0.140
	Age	0.553
	BMI	0.010
	CAD * Sex	0.321

‘*’ indicates adjustment for potential interactions between the variables ‘CAD’ and ‘Sex’ to assess the association of LAD-LCx and RCA-aorta angle and CAD between males and females. All other abbreviations are the same as in Table 1.

## Data Availability

Data is unavailable due to ethical restrictions.

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
