# Peer review of "Investigation of the Relationship between Right Coronary Artery–Aorta Angle and Coronary Artery Disease and Associated Risk Factors"

_jcm, 2023, doi:10.3390/jcm12031051_

Round 1
Reviewer 1 Report
The study by Dr Geerlings-Batt et al presents a very interesting hypothesis: that there might be a relationship between Ao/RCA angle and the development of CAD. From the data presented, I deduce that this angle may play a certain role, conditioned by local wall stress and vortex formation, but the significant overlap of data does not allow to establish an angle threshold from which a certain patient would be considered high-risk only based on angle data, without considering other risk factors.
As such I have some major comments:
1. This relatively difficult method seems to need a lot of expertise, I am not sure that it would lead to a clinical tool, maybe it will just explain some local hemodynamics playing in the atherogenesis in the coronaries
2. Previous data, and your discussion, speak over the role of aortic root dilatation in the coronary emergence angle, which is completely logical. Why did you not measure the ARD and not include it in your analysis? I would like to see data on this. ARD is a very simple measurement, and may give us the same prognostic information as the RCA/Aorta angle.
Further comments as they arise during the lecture of the manuscript:
Abstract.
Motivation: Just because the angle of the LAD and LCx has been explored does not mean that the RCA angle needs to be explored. I would rather state that at the level of the left coronary artery there is evidence regarding an association between bifurcation angle and CAD, and you wanted to explore similar associations at the level of the RCA. There is a problem in the reasoning: in the left coronary the association has not been shown for the left main, but for its bifurcation, so if you want to extrapolate the idea, it should refer to an RCA bifurcation. Therefore, motivating that the LAD/Cx angle urges us to explore the RCA angle seems to me an unfounded extension if presented so directly.
Introduction:
-In my opinion Figure 2 is not necessary, diverts the attention from the problem and even undermines your theoretical assumptions. You develop the theory behind LAD/Cx bifurcation, where one main tube splits into two relatively similar diverging tubes. This is not the case of the RCA (one large tube under high pressure with one small emerging tube with pressure drop).
-The whole paragraph on page 2 lines 51-58 can be shortened to a conclusion, with the necessary references.
-You then state on page 3 line 67-68: ”…is yet to explore…”. Why is it necessary to explore those? Why namely the RCA and not the LM? Why not both then? As explained in the comment to the abstract, you are making here a jump over some logical steps. Just knowing the evidence concerning the LAD/Cx angle does not immediately imply that we need to study the RCA angle. My advice is to complement the missing pieces in the construction of your argument.
-The additional purpose paragraph over risk factors, page 3 lines 77-85: is it really necessary? These are rather confounding factors in your analysis. They lead to CAD, regardless of the angle of the coronaries. You want to be able to take apart the role of the angle. This paragraph belongs in methods, does not need to be in introduction.
Results:
Page 5 line 164: RCA-LCx angle: probably you mean RCA-aorta?
Figure 4:
Indeed, by looking at the distributions we can see the following: For LAD/Cx the normal have mostly narrow angles, but there is significant overlap with CAD angles, even if the average values are significantly different. Further, for the RCA, significant overlap with the normal, with the average “pulled” to the left by several very low-angle patients.
Table 2: By LAD/Cx you name the CAD group “diseased”, by RCA/aorta “CAD”. Please be consistent, is it the same target endpoint/group?
Table 3-4
The correlation with BMI is very significant for RCA/Aorta. Is that not a problem for your general linear model? Risk of collinearity?
Discussion:
-You start your discussion with an old and excellent anatomical study, very well founded. The respective study rightfully supposed that aorta dilatation would play a role in the angle with the RCA, progressive dilatation leading to “rolling” of the aorta root to the right and narrowing of the RCA emergence angle.
-Unfortunately, your analysis contains no data on the aortic root dilatation/size, and its causal/confounding role in the angle, and the relation with CAD. Maybe just measuring the aorta would be sufficient to estimate CAD, making the very difficult RCA angle not necessary. You also point out that aortic root dilatation is associated with risk factors
-Pericardial fat volume should not influence the coronary tree, it builds up over it, should not push aside the coronaries. Heart size/LV size, as well as aorta position and root dilatation may be influenced by BMI. This merits further attention
Limitations:
You set in the limitations the lack of analysis for ARD. But you made no attempt to analyzing it, which remains possible at least for univariate analysis. If it turns out as very significant, you may consider removing the BMI confounder (highly correlated) from the multivariate analysis and using ARD. I am also curious about the correlation ARD with Ao/RCA angle.
Author Response
The study by Dr Geerlings-Batt et al presents a very interesting hypothesis: that there might be a relationship between Ao/RCA angle and the development of CAD. From the data presented, I deduce that this angle may play a certain role, conditioned by local wall stress and vortex formation, but the significant overlap of data does not allow to establish an angle threshold from which a certain patient would be considered high-risk only based on angle data, without considering other risk factors.
As such I have some major comments:
- This relatively difficult method seems to need a lot of expertise, I am not sure that it would lead to a clinical tool, maybe it will just explain some local hemodynamics playing in the atherogenesis in the coronaries.
Response: This is completely valid. We agree our method of measuring RCA-aorta angle is unappealing in its current state. If it were to be implemented in routine practice, the approach would undoubtedly require refinement, or perhaps the use of software to aid in efficiency. This has now been highlighted as a limitation in lines 409-413. We also suggest the relationship between RCA-aorta angle and potentially altered haemodynamics may be useful in identifying patients at greater risk of developing atherosclerosis prior to there being radiological evidence of plaque formation.
- Previous data, and your discussion, speak over the role of aortic root dilatation in the coronary emergence angle, which is completely logical. Why did you not measure the ARD and not include it in your analysis? I would like to see data on this. ARD is a very simple measurement and may give us the same prognostic information as the RCA/Aorta angle.
Response: Thank you for your suggestion. Possible associations between ARD and RCA-aorta angle were not considered until later in the study. We agree that this may provide valuable data, however, we believe this to extend beyond the scope of our study. Whilst it is an avenue for future research, it may be better explored in a dedicated study, which we have summarised in lines 402-405.
- Abstract: Motivation: Just because the angle of the LAD and LCx has been explored does not mean that the RCA angle needs to be explored. I would rather state that at the level of the left coronary artery there is evidence regarding an association between bifurcation angle and CAD, and you wanted to explore similar associations at the level of the RCA. There is a problem in the reasoning: in the left coronary the association has not been shown for the left main, but for its bifurcation, so if you want to extrapolate the idea, it should refer to an RCA bifurcation. Therefore, motivating that the LAD/Cx angle urges us to explore the RCA angle seems to me an unfounded extension if presented so directly.
Response: Thank you for your suggestion. We have revised our motivation of the study as suggested in both the abstract and Introduction.
Introduction: In my opinion Figure 2 is not necessary, diverts the attention from the problem and even undermines your theoretical assumptions. You develop the theory behind LAD/Cx bifurcation, where one main tube splits into two relatively similar diverging tubes. This is not the case of the RCA (one large tube under high pressure with one small emerging tube with pressure drop).
Response: Thank you for your suggestion. For readers who are unfamiliar with CFD analysis, this figure is intended to provide a visual representation to assist in improving understanding. Without prior experience working with CFD, it may be difficult to conceptualize, hence, Figure 2 is meant to be supplementary to the introductory paragraph discussing CFD with respect to LAD-LCx angle.
- The whole paragraph on page 2 lines 51-58 can be shortened to a conclusion, with the necessary references.
Response: Thank you for your suggestion. We have shortened this paragraph as suggested..
- You then state on page 3 line 67-68: ”…is yet to explore…”. Why is it necessary to explore those? Why namely the RCA and not the LM? Why not both then? As explained in the comment to the abstract, you are making here a jump over some logical steps. Just knowing the evidence concerning the LAD/Cx angle does not immediately imply that we need to study the RCA angle. My advice is to complement the missing pieces in the construction of your argument.
Response: We agree to reviewer’s insightful comment. We have revised the sentence to justify the rationale of this study.
- The additional purpose paragraph over risk factors, page 3 lines 77-85: is it really necessary? These are rather confounding factors in your analysis. They lead to CAD, regardless of the angle of the coronaries. You want to be able to take apart the role of the angle. This paragraph belongs in methods, does not need to be in introduction.
Response: Thank you for highlighting this. One aim of our study was to further characterize the relationship between CAD risk factors and the coronary angles. Hence, we were asking the question, “Is there an association between CAD and coronary angle? And/or its risk factors and coronary angle?” We agree, these may very well be confounding variables, but we are trying not to dismiss potential relationships based on this probability.
We understand how this paragraph may be better suited to the Methods section. However, the purpose of this paragraph is to introduce our aim, “to identify relationships between CAD risk factors; age, sex, BMI, smoking status, diabetes, hypertension, and high blood-cholesterol; and RCA-aorta and LAD-LCx angles, to further characterize the role variable coronary anatomy plays in CAD development,” and hence is positioned in the Introduction section.
- Results: Page 5 line 164: RCA-LCx angle: probably you mean RCA-aorta?
Response: Thank you very much for noticing this typo, it now reads “RCA-aorta angle”.
- Figure 4: Indeed, by looking at the distributions we can see the following: For LAD/Cx the normal have mostly narrow angles, but there is significant overlap with CAD angles, even if the average values are significantly different. Further, for the RCA, significant overlap with the normal, with the average “pulled” to the left by several very low-angle patients.
Response: This is true, there are significant overlaps between the normal and CAD curves for both angles. This may be due to the fact that coronary angle is not the only factor related to CAD, and if we assume the direction of causality to be coronary angle influencing CAD, the risk factors highlighted in this study are also affecting CAD pathogenesis. Hence, overlap is expected, and not mutually exclusive to the association between CAD and coronary angle. The idea is that coronary angle is one of many patient-specific factors we can use to ascertain a person’s risk of developing CAD.
- Table 2: By LAD/Cx you name the CAD group “diseased”, by RCA/aorta “CAD”. Please be consistent, is it the same target endpoint/group?
Response: Thank you for noticing this. Yes, this is referring to the same group. We are now more consistent and refer to this group as the “CAD” group throughout all tables.
- Table 3-4: The correlation with BMI is very significant for RCA/Aorta. Is that not a problem for your general linear model? Risk of collinearity?
Response: We agree, there is a risk of collinearity when considering BMI and CAD versus RCA-aorta angle. This is why we chose to include BMI as an independent variable in our general linear model (GLM), however, you are correct, this relationship still poses a small risk to validity. Although our GLM suggests BMI is related to RCA-aorta angle, independent of CAD, our sample size limited the number of variables we could include in our GLM. Hence, based on our results, we suggest BMI is related to RCA-aorta angle, but highlight in our discussion that future research should attempt to overcome/investigate possible collinearity.
- Discussion: You start your discussion with an old and excellent anatomical study, very well founded. The respective study rightfully supposed that aorta dilatation would play a role in the angle with the RCA, progressive dilatation leading to “rolling” of the aorta root to the right and narrowing of the RCA emergence angle. Unfortunately, your analysis contains no data on the aortic root dilatation/size, and its causal/confounding role in the angle, and the relation with CAD. Maybe just measuring the aorta would be sufficient to estimate CAD, making the very difficult RCA angle not necessary. You also point out that aortic root dilatation is associated with risk factors.
Response: Thank you for your comments. We agree, this seems to be a promising direction for further research, and you may be correct, ARD may be a more reliable/valid indicator of a person’s risk of developing CAD than RCA-aorta angle. However, investigating ARD was not the purpose of our study, although it has been highlighted as an area for future exploration in lines lines 402-405.
- Pericardial fat volume should not influence the coronary tree, it builds up over it, should not push aside the coronaries. Heart size/LV size, as well as aorta position and root dilatation may be influenced by BMI. This merits further attention.
Response: We thank the reviewer for insightful comments. Paragraph 4.4 “LAD-LCx Angle and CAD Risk Factors” has been revised regarding the discussion around pericardial fat and LAD-LCx angle. Thank you for your comment. Although relative pericardial fat volume should not expand coronary angles, there appears to be no literature discussing this relationship, or lack of relationship. We suggest relationships between heart size, ARD, BMI and coronary angles be further investigated.
- Limitations: You set in the limitations the lack of analysis for ARD. But you made no attempt to analyzing it, which remains possible at least for univariate analysis. If it turns out as very significant, you may consider removing the BMI confounder (highly correlated) from the multivariate analysis and using ARD. I am also curious about the correlation ARD with Ao/RCA angle.
Response: See our responses to your comments 2 and 12.
Reviewer 2 Report
The paper can be interesting, but the results appear very weak. Furthermore the reported series of patients presents many selection bias. This makes the results very debatable and inconclusive. The Authors should try to aknowledge and discuss in more detail all the limitations of the study, in order to help the reader to understand difficulties in methodology and perspectives, to possibly utilize these results to plan future research.
For the pre-selection bias to be discussed see lines 88-102. It should also be clearly stated that patients were not consecutive. The indications for CCTA should also be more clearly reported, in more detail than in line 88.
Line 105: and Type 1 diabetes?
LIne 106: please report the definition of high blood cholesterol.
Line 129-131: it is not clear why not using 3D reconstruction to identify 2D section where to measure the angles. Perhaps a more detailed discussion and comparison, referring also to methodology details in Reference 4 could be helpful. Why not separating patients in three groups (normals, mild plaques, severe plaques) as in Reference 4? Which was the adopted definition of plaque presence ? and the distinction with minimal plaques? How patients with only mild localized increased thickness of coronary arteries' wall were classified? Which was the cut-off for plaque definition ?
For the statistical analysis and significance, why ANOVA between and within groups was not used?
Line 156: the abbreviation for "general linear model" (GLM) should be added in the text, since later on, in the paper, we can read the abbreviation GLM alone, difficult to understand (see lines 307 and 309).
In the results it is not clear how much lenght and size (diameters) of ascending aorta could influence coronary angle measurements. The same for the angle between the long axis of the LV and the emerging aorta, This angle should become more acute with increasing lenght of the aorta, due to atherosclerotic changes, and could also influence the coronary angle measurements. All these data sould be also measured and compared.
In Table 2 the results of ANOVA statistical analysis for multiple comparisons between and within groups should be reported. In the discussion and limitations it should be reported in detail the weakness of the results, which can not be used fruitfully in the individual subjects , due to the wide overalap of SD.
Table 3. The correlation with BMI could be better related to height than weight component of the patients. Please check correlation also separately with weight and height.
In the discussion a tentative explanation shoud be offered why for the right coronary artery the correlation with CAD is with a narrow acute angle and in the left coronary artery with a larger angle ?
Lines 231-233: not clear: please explain, also with references.
Lines 289-291: not clear: please explain
Author Response
- The paper can be interesting, but the results appear very weak. Furthermore the reported series of patients presents many selection bias. This makes the results very debatable and inconclusive. The Authors should try to acknowledge and discuss in more detail all the limitations of the study, in order to help the reader to understand difficulties in methodology and perspectives, to possibly utilize these results to plan future research. For the pre-selection bias to be discussed see lines 88-102.
Response: Thank you. You are correct. Our study contains inherent biases resulting from our sampling technique. Lines 416-429 have been included to acknowledge the challenges that were associated with our methodology. We have also suggested future research include incomplete data with more comprehensive data analyses to overcome biases associated with missing data.
- It should also be clearly stated that patients were not consecutive. The indications for CCTA should also be more clearly reported, in more detail than in line 88.
Response: Thank you for your suggestion. We have revised lines 98-100 to explain that patients with suspected CAD were those “presenting with angina pectoris, with or without a family history of CAD, and/or those with abnormal echocardiogram or electrocardiogram results”.
- Line 105: and Type 1 diabetes?
Response: Type 1 diabetes was not included in this study, since only information regarding Type 2 diabetes is routinely collected prior to CCTA.
- Line 106: Please report the definition of high blood cholesterol.
Response: Blood cholesterol is reported as a categorical variable (Y/N) in our study, since this is how patient information was collected by the clinical site prior to CCTA. Patients were simply asked to report whether they had high blood-cholesterol. We recognise that this also introduces a self-reporting bias, which has been acknowledged in lines 421-429.
- Line 129-131: it is not clear why not using 3D reconstruction to identify 2D section where to measure the angles. Perhaps a more detailed discussion and comparison, referring also to methodology details in Reference 4 could be helpful.
Response: Thank you for your suggestion. These details have been summarised in lines 143-153. Reference 4 found measuring LAD-LCx angle via axial images to be of comparable accuracy to 3D views, and Reference 18 found using MPRs to be the more accurate method for measuring RCA-aorta angle compared to using axials alone.
- Why not separating patients in three groups (normals, mild plaques, severe plaques) as in Reference 4? Which was the adopted definition of plaque presence? and the distinction with minimal plaques? How patients with only mild localized increased thickness of coronary arteries' wall were classified? Which was the cut-off for plaque definition?
Response: This is a great suggestion. Unfortunately, this would have required each patient to undergo invasive coronary angiography (ICA) to confirm degree of coronary stenosis, which they did not in our study. Instead, patients were either characterised as having CAD, or not having CAD, based on CCTA appearance. This has been discussed in lines 324-329 and lines 404-408.
- For the statistical analysis and significance, why ANOVA between and within groups was not used?
Response: This is a good idea for future research investigating disease severity, such as in those that may subdivide groups according to normal vs mild vs severe plaques. However, since our study did not employ the use of ICA to characterise degree of stenosis, we did not subdivide the diseased group and consequently, did not use ANOVA for subgroup analysis. This has been discussed at lines 324-329 and lines 404-408.
- Line 156: the abbreviation for "general linear model" (GLM) should be added in the text, since later on, in the paper, we can read the abbreviation GLM alone, difficult to understand (see lines 307 and 309).
Response: Thank you for your suggestion. The abbreviation for general linear model has now been provided at line 173.
- In the results it is not clear how much length and size (diameters) of ascending aorta could influence coronary angle measurements. The same for the angle between the long axis of the LV and the emerging aorta, This angle should become more acute with increasing length of the aorta, due to atherosclerotic changes, and could also influence the coronary angle measurements. All these data should be also measured and compared.
Response: We thank the reviewer for constructive suggestion the results. Yes, the length and size of the ascending aorta could influence coronary angle measurements although this was not addressed in previous studies and our current study. We have highlighted it in the study limitations as part of future research.
- In Table 2 the results of ANOVA statistical analysis for multiple comparisons between and within groups should be reported. In the discussion and limitations it should be reported in detail the weakness of the results, which cannot be used fruitfully in the individual subjects, due to the wide overlap of SD.
Response: We were unable to use ANOVA in this way as it would require independence of data, whereas these groups consisted of the same patients. A two-way ANOVA could have been used; however the sample size was too small to achieve statistical significance.
- Table 3. The correlation with BMI could be better related to height than weight component of the patients. Please check correlation also separately with weight and height.
Response: Thank you for your comments. This may be true. As per the above suggestion, height, and weight versus LAD-LCx and RCA-aorta angle have been correlated. According to our data, height and weight were both weakly-moderately positively correlated with LAD-LCx angle, and weight was moderately negatively correlated with RCA-aorta angle. There was no significant correlation identified between RCA-aorta angle and height. The purpose of using BMI was to standardise patient weight with respect to body habitus. I.e. whether a person is overweight at 70kg is dependent on their height etc. Hence our predefined CAD risk factor was BMI, specifically.
- In the discussion a tentative explanation should be offered why for the right coronary artery the correlation with CAD is with a narrow acute angle and in the left coronary artery with a larger angle?
Response: Thank you for highlighting this gap in our discussion. A tentative explanation has been provided at line 309-311.
- Lines 231-233: not clear: please explain, also with references.
Response: Explanation is provided with citation of some references to support our statement/argument as shown at lines 315-317.
- Lines 289-291: not clear: please explain
Response: We have explained it with citation of recent studies as shown at lines 377-381.
Round 2
Reviewer 1 Report
The authors addressed my comments in a satisfactorily manner.
Reviewer 2 Report
The Authors made the required changes.